# The Distribution of Available Prevention and Management Interventions for Fetal Alcohol Spectrum Disorder (2007 to 2017): Implications for Collaborative Actions

**DOI:** 10.3390/ijerph16122244

**Published:** 2019-06-25

**Authors:** Babatope O. Adebiyi, Ferdinand C. Mukumbang, Charlene Erasmus

**Affiliations:** 1School of Public Health, University of the Western Cape, Cape Town 8001, South Africa; mukumbang@gmail.com; 2Child and Family Studies, University of the Western Cape, Cape Town 8001, South Africa; cjerasmus@uwc.ac.za

**Keywords:** fetal alcohol spectrum disorder, perinatal alcohol exposure, prevention, management, interventions, scoping review, pregnant women, children, adults, development disabilities

## Abstract

The global prevalence of Fetal Alcohol Spectrum Disorder (FASD) remains high despite the various preventive and management interventions that have been designed and implemented to tackle the issue in various settings. The aim of the scoping review is to identify and classify prevention and management interventions of FASD reported globally across the life span and to map the concentration of these interventions across the globe. We searched some selected databases with predefined terms. Framework and narrative approaches were used to synthesize and report on the findings. Thirty-two prevention intervention studies and 41 management interventions studies were identified. All the interventions were reported to be effective or showed promising outcomes for the prevention and management of FASD, except four. Although Europe and Africa have a relatively higher prevalence of FASD, the lowest number of interventions to address FASD were identified in these regions. Most of the interventions for FASD were reported in North America with comparatively lower FASD prevalence. The uneven distribution of interventions designed for FASD vis-à-vis the burden of FASD in the different regions calls for a concerted effort for knowledge and intervention sharing to enhance the design of contextually sensitive preventive and management policy in the different regions.

## 1. Introduction

Fetal alcohol spectrum disorder (FASD) is a diagnostic term used to describe a range of conditions affecting individuals exposed to alcohol prenatally [1]. FASD can manifest as fetal alcohol syndrome (FAS), partial FAS, alcohol-related neurodevelopmental disorder (ARND) and alcohol-related birth defects (ARBD) [2,3]. FASD may result in primary and secondary disabilities for individuals prenatally exposed to alcohol. Primary disabilities may include abnormal facial features, learning disabilities, attention difficulties, poor memory, poor reasoning and judgment skills, hyperactive behavior and poor coordination [4]. Without appropriate interventions to mitigate primary disabilities, secondary disabilities may develop. Secondary disabilities may include mental health problems, disrupted school experience, trouble with the law, confinement, inappropriate sexual behavior and alcohol/drug problems [5]. 

One in 13 alcohol-exposed pregnancies results in diagnosable FASD and the global prevalence of FASD is estimated at eight per 1000 children and youth [6]. A systematic review using 2012 data estimated the prevalence of FASD around the world among children and youth in general population using World Health Organization (WHO) regions [6] as shown in the chart below (Figure 1). 

The World Health Organization (WHO) recognizes FASD as a public health issue and has developed a guideline for the identification and management of substance use and substance use disorders in pregnancy [7]. The guideline recommends that health professionals should ask all pregnant women about their use of alcohol and brief intervention should be given to all pregnant women using alcohol. The Center for Disease Control and Prevention (CDC) has also identified and supported the implementation of two interventions in preventing and reducing the prevalence of FASD [8]. One of the interventions is alcohol screening and brief intervention (SBI)—an intervention that identifies and helps individuals who are drinking excessively. The second intervention is the CHOICES program—an evidence-based intervention that increases motivation and commitment to reduce or stop drinking and/or use contraception effectively [8].

In addition to the efforts of WHO and CDC, researchers have developed and implemented various interventions such as universal prevention, brief intervention, motivational interview, case management, and service provider training for the prevention of FASD [9,10,11,12,13]. These interventions have been found effective or have shown promising outcomes in preventing FASD and/or reducing the risk of prenatal alcohol exposure [9,10,11,12,13]. Although, the conclusions drawn from the reviews were limited by the poor methodological quality and paucity of studies as indicated by the authors [14,15]. Psychological and educational interventions and other interventions delivered during antennal care have potentials to increase abstinence and reduce alcohol consumption among pregnant women [14,15]. On the other hand, various barriers have been found to mitigate the effectiveness of these interventions. These barriers may include conflicting messages in the media, stigmatization of people with alcohol use disorders and birth mothers, fear of losing child custody (which may prevent women from seeking help) and lack of skills by service providers to discuss alcohol use with women who are pregnant [16]. The challenges regarding the prevention of prenatal alcohol exposure during pregnancy and unplanned pregnancy through the use of contraceptives necessitate interventions to manage primary disabilities and prevent or minimize the impacts of secondary disabilities. 

Pei et al. [17] and Jirikowic et al. [18] suggested that because the needs of individuals with FASD are diverse, their assessment and management should be personalized. Most importantly, early diagnosis and intervention could minimize the impact of some of the problems that arise from prenatal alcohol exposure [19,20,21,22,23,24]. Reid et al. [22] showed that there is growing evidence supporting the effectiveness of FASD management interventions in improving outcomes for early to middle childhood. However, there is a dearth of management interventions for individuals with FASD beyond early and middle childhood [22] suggesting inadequate management plans for these individuals. Peadon et al. [23] reported that pharmacological and non-pharmacological management interventions have shown some benefit among children with FASD. While Paley and O’Connor [25] confirmed that behavioral interventions showed immediate post-intervention effects, they did not ascertain the long-term follow-up outcomes.

Systematic reviews have been published on interventions for the prevention [14,15] and management of FASD [22,23] and other literature reviews for management interventions [25,26,27,28,29]. The systematic reviews on the prevention of FASD focused mainly on interventions for pregnant women, women planning a pregnancy and indigenous community without considering women in other groups [14,15,30]. One of the systematic reviews for management interventions includes both pharmacological and non-pharmacological interventions for children with FASD only [23]. The second systematic review on management interventions for FASD examined the effectiveness of interventions across the life span, however, only non-pharmacological interventions were included [22]. From the reviews above, we observed that there is a lack of a review for prevention of FASD outside those who are pregnant, planning a pregnancy and indigenous community. In addition, we discovered the lack of review for both pharmacological and non-pharmacological interventions that assessed methodological rigor across the life span. FASD interventions need to be diverse and include prevention and both pharmacological and non-pharmacological management approaches across the life span as the negative effects of prenatal alcohol exposure can manifest across all ages. Therefore, the need to address the gap of a study exploring the prevention interventions for all women and the management interventions across life span necessitates this current scoping review.

In addition to the above motives for this scoping review, Premji et al. [31], in their study on research-based interventions for individuals with FASD, found weak scientific evidence for the effectiveness of interventions across the life span. Therefore, they called on researchers, service providers, and policymakers to collaborate and develop appropriate interventions for individuals with FASD. For this to be achieved, knowledge of available interventions for the prevention and management of FASD and the assessment of their effectiveness should be known. 

To this end, we aimed to conduct a scoping review that will serve four purposes. First, the review will help to identify the prevention and management interventions of FASD reported in the literature across the life span to address the above-mentioned gaps in interventions for FASD. Secondly, we sought to map the concentration of these interventions across the globe vis-à-vis the burden of FASD in the different regions. Thirdly, to update previous reviews and lastly to provide current information to help inform policy development [32,33,34,35].

## 2. Materials and Methods

We conducted a scoping review to identify and classify the prevention and management interventions of FASD reported in the literature across the life span that may be included in the policy for FASD. We adapted five steps for conducting reviews as proposed by Arksey and O’Malley [36] to suit the aim of this study. The five steps include (1) framing of questions for the review; (2) identifying relevant work; (3) assessing the quality of studies; (4) summarizing the evidence; (5) interpreting the findings.


*Step 1: Framing of questions for the review*


We formulated the review question using the PICO mnemonics (Population, Intervention, Comparison, and Outcome) (Table 1) [37]. What are the prevention and management interventions of FASD available globally? 


*Step 2: Identifying relevant work*


We searched the following Ebsco Host embedded databases; Academic Search Complete, ERIC, SoINDEX, Health Source: Nursing/Academic Edition, CINAHL, Medline and Psych-ARTICLES. In addition, we searched Sabinet, SAGE Journals, and PubMed databases. The following standard Boolean phrase was applied to all the databases: (fetal alcohol spectrum disorder” OR “fetal alcohol syndrome” OR “alcohol-related neurodevelopmental disorder” OR “alcohol-related birth defects” OR “partial fetal alcohol syndrome” OR “prenatal alcohol exposure”) AND (“intervention” OR “strategy” OR “treatment” OR program” OR “management” OR “prevention” OR “therapy”). From the results of the search, the titles and abstracts of the articles were screened by two of the authors (B.O.A. and C.E.) for possible inclusion using the following criteria.

Inclusion criteria:The interventions (both pharmacological and non-pharmacological) must aim at preventing or improving the outcome of prenatal exposure to alcohol;Articles published in the English Language;Articles published from 2007 to 2017 (we chose this period to provide current information to help inform policy development);The target population must be women, young people, and individuals with FASD;All the types (randomized controlled trials (RCT), quasi RCT, non-randomized controlled trials and cohort studies with pre- and post-intervention);Interventions targeting any age group.

Exclusion criteria:Studies that do not report on the effectiveness or promising outcome of the interventions;Animal studies;Other systematic, scoping and literature reviews;Unpublished prevention and management interventions;Articles published before 2007 and after 2017.

The flow chart of the selection process is presented in Figure 2.

The database search yielded 2814 articles and 2495 were included after removal of the duplicates. Eight-two articles were potentially relevant articles after the title and abstract screening and 73 after the retrieval of the full text. Seventy-three articles met the inclusion criteria for the final synthesis. B.O.A. and F.C.M. independently searched and screened the included articles and all the disagreements were resolved by C.E. (see Figure 2).

This study was approved by the research ethics committee of the University of the Western Cape (BM/16/4/4).


*Step 3: Accessing the quality of studies*


We used a study design-based quality checklist: The Effective Public Health Practice Project (EPHPP) assessment tool. This tool was developed to assess the methodological rigor of primary studies in public health [38]. The EPHPP assessment tool comprises of six quality components: selection bias, study design, confounders, blinding, data collection methods, and withdrawals and dropouts. Each study was rated using the “strong,” “moderate,” or “weak” scale (see Table 2). Herein the overall rating of the quality of the studies assessed was not done. Additionally, the rating was not used to determine the studies to be included in this review. This is because Juni et al. [39] recommended that studies should be assessed independently and the total score should not be used. This recommendation was supported by other authors [22]. The importance of the assessment is to provide support to the findings and conclusion drawn from this study. The findings from this study should interpreted with a due consideration to the assessment. B.O.A. and C.E. independently assessed the included articles and all the disagreements were resolved by F.C.M. (see Appendix A).

The qualities of the studies were assessed, and only two studies were rated “strong” for selection bias. Studies were rated “strong” if the participants were randomly selected from the target population, “moderate” if the participants were recruited from a clinic and “weak” if the participants were self-referred. Forty studies were rated “strong” for study design, which represents the number of randomized controlled trials that were included. Studies were rated “strong” if the studies were randomized controlled trials or controlled clinical trials “moderate” if the studies were cohort analytic, case-control, cohort, or an interrupted time series and “weak” if the studies were other designs or designs not stated. Sixty-six studies were rated “strong” for data collection. Studies were rated “strong” if the tools were valid and reliable “moderate” if the tools were valid but had not been shown to be reliable and “weak” if the tools were not reliable or valid. Appendix A contains the details of the quality assessments of the studies included. 


*Step 4: Summarizing the evidence*


We summarized the data by tabulating the study characteristics, intervention approach and results of the studies (see Appendix A). We employed a type of narrative synthesis [40] called thematic summaries. Thematic summaries allow for categorization of studies into thematic groups and tabulation of the findings into a thematic framework based on the predefined categories (framework) [41]. Therefore, we used the framework method [42] to develop themes for data analysis (see Figure 3).

## 3. Results

The characteristics of the studies included in the review are illustrated in Table 3.

### 3.1. Global Distribution of FASD Prevention and Management Interventions

We mapped the distribution of the prevention and management interventions based on settings within which each of the identified studies were conducted (Figure 4). Most of the studies were conducted in the United States (47 of the 73 studies). Canada recorded the second highest studies on FASD prevention and management studies (14 studies). Five studies where identified to be conducted in South Africa and three from Australia. Ukraine, Poland, Spain, and Japan had one study each. Our analysis showed no other published articles in any other countries.

### 3.2. Prevention Interventions 

Prevention interventions were identified as interventions aimed at either preventing or reducing the prevalence of FASD. We classified these interventions as facilities-based, school-based/education-based and community-based depending on where the interventions had been carried out (Table 4). However, interventions that took place in more than one setting were classified in the one that seemed to be most appropriate for easy presentation of findings.

#### 3.2.1. Facility-Based Prevention Interventions

The review identified 10 facility-based interventions [13,43,44,45,46,47,48,49,50,51]. Eight of the facility-based interventions [43,44,46,47,48,49,50,51] targeted women while the other two were aimed at service providers [13,45]. We identified three motivational interviews [43,50,51] (single-session motivational interviews [43,50] and a motivational interview [51]). One of the interviews decreased alcohol consumption during pregnancy [43] whereas the remaining two were not found effective in decreasing alcohol use [50,51]. Another study described the effectiveness of the use of a dual-focused approach (motivational interviewing and the trans theoretical model) in reducing the risk for an alcohol-exposed pregnancy (AEP) [44]. In addition, Project CHOICES intervention for women decreased the AEP risk through the effective use of contraception and decreased alcohol use [49]. We found a web-based alcohol assessment and personalized feedback for women of reproductive age to reduce the amount of risky alcohol consumption [47]. We identified a short training course based on brief motivational interviewing principles to build service provider capacity for the better management of women at risk for AEP [13]. Another study described the distribution of educational resources to increase health professionals’ knowledge, change health professionals’ attitudes’ and practice on FASD and improve the quality of advice the health professionals provide to pregnant women vis-à-vis alcohol consumption [45]. We also found two brief interventions for pregnant women [46,48] (Computer-Delivered Screening and Brief Intervention and Brief Computer-Delivered Intervention). The first intervention demonstrated efficacy for favorable birth outcome [46] whereas the other significantly decreased alcohol use [48].

#### 3.2.2. School-Based/Education-Based Prevention Interventions

Three school-based interventions [52,53,54] were identified and reported to have the potential of preventing FASD. We found a peer-led 40-minute pilot multimedia presentation intervention among middle and high school students that increased the students’ knowledge about the effects of alcohol consumption during pregnancy [52]. We also found multimedia, peer-delivered educational presentation for youth (Fetal Alcohol Spectrum Teaching and Research Awareness Campaign) improved youths’ knowledge about FASD [53]. In addition, the third education-based intervention, educational leaflet improved women’s knowledge of FASD [54].

#### 3.2.3. Community-Based Prevention Interventions 

Nineteen community-based interventions [9,10,11,12,55,56,57,58,59,60,61,62,63,64,65,66,67,68,69] aimed at either preventing FASD or reducing the prevalence of FASD were identified. We found three CHOICE programs [55,57,69]. The first program (the Oglala Sioux Tribe CHOICES) is the motivational interview techniques based pre-conceptional prevention of AEP and the program reduced risky drinking in women at risk for AEP and/or preventing unintended pregnancy [55]. The second program (the Project Healthy CHOICES) is a self-administered, mail-based prevention intervention and it was found effective in minimizing the risk of AEP [57]. The third Project CHOICES intervention for youth and adult demonstrated increased effectiveness of birth control use and decreased use and abuse of alcohol [69]. Three case management prevention interventions [12,60,67] were identified. Two case management intervention (intervention activities that incorporated life management, Motivational Interviewing techniques and the Community Reinforcement Approach (CRA)) that helped women to stop drinking or drink less while pregnant, thereby decreasing the risk of FASD [12,67]. The third case management intervention study was found effective in reducing maternal drinking at critical times and therefore decreased the fetal alcohol exposure level [60].

We identified four brief interventions [10,59,62,64] aimed at either preventing FASD or reducing the prevalence of FASD. We found a brief intervention with a potential of promoting abstinence to alcohol among women [10]. In another study, the Targeted Screening, Brief Intervention, and Referral to Treatment (SBIRT) intervention decreased risky drinking behavior and vulnerability to AEP among women [59]. A different study found a telephone-based brief intervention to be successful and cost-effective in reducing the risk of AEP in women [62]. The last study reported brief motivational intervention to be effective in minimizing the risk of AEP in women [64].

Only two web-based interventions [11,58] were identified. In the first intervention, the risk of AEP was diminished using tailored motivational messages [58]. The second intervention was based on a motivational interview in which participants received intervention materials through email was effective in modifying self-reported drinking and contraception behavior [11].

Our review also identified a training program [65], media campaign [66] and parent program based in the community [63]. We found an alcohol-server training program on responsible beverage service to be effective in reducing the serving of alcohol to visibly pregnant women [65]. Another study reported that a media campaign (including posters, radio ads, and other materials such as brochures and pens) decreased risky drinking behavior among women [66]. The First Step Program (a mentorship program model after the Parent-Child Assistance Program) demonstrated promising outcomes for women at-risk for giving birth to a child with FASD [63]. In addition, we identified pre-conceptional motivational interviewing interventions for women to decrease DDD (drinks per drinking day), ineffective contraception rate, and AEP risk [68].

We also identified a combination of two threat concepts and one positive concept based on self-efficacy with promising potentials to promote women’s intentions to abstain from alcohol during pregnancy [61]. In another study, the risk of AEP was reduced in the community by one-session, remote-delivered, preconception, motivational interviewing-based intervention for non-treatment-seeking community women [56]. In addition, it was reported in a study that universal intervention can reduce the prevalence of FASD, especially where knowledge of harms of maternal drinking is low [9].

### 3.3. Management Interventions 

In a similar manner, we classified the management interventions as facilities-based, school-based/education-based and community-based depending on where the interventions were carried out (Table 5). However, interventions that took place in more than one setting were classified in the one that seemed to be most appropriate for easy presentation of findings.

### 3.4. Facility-Based Management Interventions

We identified 13 facility-based management interventions [70,71,72,73,74,76,77,78,79,80,81,82] aimed at managing the FASD. Five of the facility-based management interventions were pharmacological interventions [70,72,73,76,82]. The four studies discussed the use of choline supplements to improve cognitive performance [70], neurocognitive functioning, particularly hippocampal-dependent memory [72], tolerability for FASD management [73] and brain development [76] in children with FASD. Three of the interventions showed promising outcomes while choline supplement was not effective in improving memory, executive function and attention deficits in children with FASD. The last study reported that attention-deficit/hyperactivity disorder (ADHD) in children with FASD may be less responsive to ADHD medication [82].

Four of the facility-based management interventions improved a certain aspect of brain functions [74,78,79,81]. The use of sensory integration (SI) therapy [74] showed high efficacy on gross motor function and individual designated therapy room [78] (a room with varieties of therapy materials for the Alert Program for Self-Regulation) demonstrated effectiveness in correcting the executive function disabilities in children with FASD. In addition, sensory integration and cognitive behavioral training improved self-regulation skills and brain development [79] and the Alert® Program (TherapyWorks Inc., Albuquerque, New Mexico, USA) for Self-Regulation improved functional integrity in the neural circuitry for behavioral regulation [81].

We also identified another study that demonstrated the potential usefulness of neuro-developmentally informed intervention in a real-world setting with young children with FASD in improving developmental deficit in several domains [71]. In another study, the use of an early intervention program for drug and alcohol addicted mothers and their young children (Breaking the Cycle) was effective to mitigate some of the well-described damages caused by heavy in utero alcohol exposure [75]. A different study reported that verbal behavior treatment program (applied behavior analysis) resulted in rapid skill acquisition across several areas of functioning (communication and functional skills) in a child with FASD [77]. Furthermore, physical activity program for children with FASD showed differences in cortisol levels in children with FASD compared to Controls [80].

### 3.5. School-Based/Education-Based Management Interventions

Our review identified 16 school/education-based management interventions [83,84,85,86,87,88,89,90,91,92,93,96,97,98]. Two of the above-mentioned management interventions aimed at improving mathematical skills and behavior in children with FASD [83,87]. Both interventions were found effective in improving mathematical skills and the behavior of children with FASD. 

We identified four game/computer interventions [88,94,95,96] as a part of the school-based management interventions. The three interventions include the use of a computerized instruction for children with FASD consistent with their parent training, a game-based process and a serious game designed to teach a metacognitive control strategy in a computer game environment for children with FASD. The three interventions verified to be useful in improving self-regulation [88], cognitive [94] and reducing disruptive behaviors [95]. In the last game intervention, the virtual reality game of fire safety and street safety, children showed significantly better knowledge of the game to which they were exposed [96]. 

Three pieces of training were identified as a part of the education-based interventions [84,89,97]. The computerized attention training [84] and the Sensorimotor Training to Affect Balance, Engagement, and Learning (STABEL) (an intervention designed to train sensory control during balance) [89] and rehearsal training on working memory span of children [97]. The pieces of training were effective in improving cognitive performance [84] increasing postural sway velocity [89] in children with FASD and improving the digit span in children [97]. Additionally, we also found a program effective for the professional development of teachers [98].

We identified two social skill interventions [90,91] as a part of education-based management interventions. The social skills intervention for social information-processing among individuals with FASD [90] and a 12-session, social-skills and play-therapy outpatient treatment for elementary school-aged children with FASD and their parents [91] (the Children’s Friendship Training (CFT)) [91]. The former improved social information-processing while the latter improved social skills and led to a reduction in problem behaviors in children with FASD.

We identified an intervention developed to reduce alcohol consumption and alcohol-related negative outcomes among adolescents with FASD [93], a highly motivating modality intervention [85], the classroom language and literacy intervention [86] and group therapy for foster and adoptive caregivers and their children affected with FASD [68]. These interventions reduced the alcohol use and prevented secondary disabilities among adolescents with FASD [70], improved sensory adaptation, balance and motor performance [85], cognitive [86], executive functioning and emotional problem-solving [92] in children with FASD respectively.

### 3.6. Community-Based Management Interventions

Twelve community-based management interventions [99,100,101,102,103,104,105,106,107,108,109,110] aimed at managing the FASD were found. Two of the community-based management interventions studies discussed the use of a neurocognitive [104] and community translation [105] Math Interactive Learning Experience (MILE) program (Claire and Coles, Atlanta, Georgia, USA) to improve mathematics skills in children with FASD. Both interventions improved mathematics skills in children with FASD. 

Three of the community-based management interventions [99,102,103] identified focused on the family. These family-based interventions include support, education, advocacy and refer families to available services [99], a home-based program for high-risk and vulnerable families [102] and a program targeting key risk and protective factors for children with FASD and their families [103]. These interventions were effective in assisting families of children with FASD [99], improving children with FASD’s self-regulatory skills by focusing on the parent-child relationship [102] and preventing secondary conditions and improving family adaptation [103]. 

We also identified a 12-session, social-skills and play-therapy outpatient treatment for elementary school-aged children with FASD and their parents [100], emotional understanding intervention [101], an online training on FASD for Court Appointed Special Advocates (CASA) [106] and community home-based attachment intervention for caregivers of children with FASD [107]. These interventions improved knowledge of appropriate social skills, self-concept and parent-reported social skills in children with FASD [100], self-regulation in children with FASD and caregiver behavior [101], the CASA workers knowledge of FASD [106] and children’s ability to communicate their needs better [107].

Furthermore, we identified three other community-based management interventions [108,109,110]. The caregiver education and training for behavioral regulation improved caregivers’ knowledge of FASD [108]. Another intervention reported is the Step by Step mentor program for parents [110]. The intervention significantly reduced the client’s needs and significantly increased the client’s goals [110]. The last of the three interventions was conducted to enhance practice regarding child welfare and improve placement stability [109]. The result of the intervention showed a significant decline in the number of placement changes [109]. 

## 4. Discussion

Seventy-three studies, of which 32 were prevention interventions studies and 41 management interventions, were included in this review. We found the interventions (prevention and management) effective or to show promising outcomes in either preventing or managing FASD. Although, four of the interventions that met the inclusion criteria and were classified as facility-based interventions (two prevention and two management) were not effective because they require further research to determine their effectiveness.

Although, this review targeted interventions across the life span for both prevention and management interventions, we could not find studies for management interventions that include participants above 18 years of age for this period (2007–2017). Most of the management interventions reported targeted children. These findings aligned with the outcome of the recent systematic review, which reported the paucity of the management interventions across the life span, especially beyond childhood [22]. The demerit of the lack of interventions beyond childhood is that secondary disabilities can more easily manifest. This is because the issues of FASD are life long and multifaceted (containing educational, social and medical), which can manifest in primary and secondary disabilities [5]. Therefore, there is a need to develop multifaceted interventions—as FASD affects individuals across the life span [24]—and management interventions for the management of secondary disabilities in adolescents and adults with FASD. This is because it has been reported that early and age/culturally-appropriate interventions showed promising outcomes for the management of FASD [22]. Furthermore, we encourage the publication of ongoing efforts to prevent and manage FASD across the life span. 

Despite calls from researchers for urgent needs for both prevention and management interventions [31,111], they remain scant in regions with high prevalence [6]. In this scoping review, most of the prevention and management intervention studies were conducted in America, precisely the United States of America (see Table 3) with a comparatively lower prevalence of FASD compared to Europe. A total of six (three each) prevention and management interventions studies were conducted in Australia and Europe, although Europe has an overwhelmingly high prevalence of FASD (20/1000 children and youth in the general population) [6]. We found out that only five prevention and management interventions studies were conducted in Africa. These interventions studies were conducted only in South Africa. This huge uneven distribution of intervention has public health implications for the prevention and management of FASD, particularly in settings like Europe and Africa. This could further demonstrate the reason for the increasing prevalence of FASD [6].

Motivational interviewing was the most reported intervention adopted for preventing FASD in this study. It was effective in reducing alcohol consumption during pregnancy by empowering participants to make the necessary behavioral changes [11,12,13,43,44,50,51,55,56,60]. Although it was not effective in two studies, a different study has found motivational interviewing to be an effective intervention in the addiction field [112]. The use of motivational interviewing has seen an increase in health-promoting behaviors such as oral health, safe sexual practices, diet modification and physical exercise [113,114]. In addition to motivational interviewing, the brief intervention was similarly found effective in preventing FASD [10,46,48,59,62,64]. The CDC has also found Alcohol Screening and Brief Intervention effective in reducing alcohol use for all adults, including pregnant women in the medical and other settings [8].

The case management approach to preventing FASD was also found effective in reducing alcohol consumption during pregnancy [12,60,67]. The case management activities incorporated life management, Motivational Interviewing techniques and the Community Reinforcement Approach. The finding supports the call for integrated care in the prevention and management of chronic conditions [115]. In addition to case management, the training and education of relevant service providers and individuals with FASD were found effective in the prevention of FASD [9,13,52,53,65,66]. These findings support the previous study that found training and education to be effective for health promotional activities in achieving behavioral changes [116].

We found management interventions such as social skills interventions [90,91,100], school-based interventions [83,84,85,86,87,88,89,90,91,92,93] and family-centered interventions [91,99,100,102,103] effective for the management of FASD. Social skills interventions were effective in managing information-processing deficits in individuals with FASD [90,91,100]. Studies have previously reported social skills interventions to be effective in managing other conditions with behavioral problems such as Autism Spectrum Disorder, schizophrenia and Down syndrome [117,118,119]. The use of school-based interventions was also found effective in improving specific educational skills, cognitive and motor performance, social skills and regulation of behavior [59,60,61,62,63,64,65,68,69,70,71]. The school-based interventions have been found effective in managing a similar disorder such as attention deficit hyperactivity disorders [120]. In addition, family-centered interventions were found to be effective in reducing problem behavior and improving social skills [91,99,100,102,103]. The effectiveness of family-centered intervention in the management of problem behaviors in children and adolescents has been earlier reported in another study [121].

Prevention and management interventions that have been developed, implemented and evaluated and found effective in preventing and/or managing FASD in a particular setting can be modified for implementation in other settings, especially settings that have high prevalence such as Europe and Africa. Nevertheless, attention should be paid to address those context-related differences, which may impact on the effectiveness of the interventions. Therefore, various context-specific adaptations should be considered during implementation to ensure the effectiveness of the interventions. Our finding aligns with that of Petrenko and Alto [27] on the need to address the barriers to the implementation of FASD interventions, especially cultural barriers as highlighted in their work. We support the call for collaboration among researchers, cultural experts, and local stakeholders to facilitate implementation, which will enhance the effectiveness of interventions and promotes sustainability. We agree with their suggestion on the use of purveyors as cultural liaisons between researchers and local stakeholders to facilitate buy-in from the latter and the community at large. The importance of the above is to encourage tailored interventions to maximize efficacy. In addition, we believe that interventions work best when the motivations and methods reflect the communities’ world view and culture. 

In addition, the uneven distribution of interventions for the prevention and management of FASD calls for collaborative actions between researchers, clinicians, service providers and community members across the globe. Improving both the prevention and management of FASD requires evidence-based approaches to community-based and clinic-based service delivery that can overcome the geographic, regional and cultural diversity contexts in which women become pregnant. Although FASD seems to be a priority for communities and governments in America, especially the United States of America, research capacity has not been available to support the development of the context-specific knowledge needed to inform policy and practice in other regions such as Europe and Africa that have a high prevalence of FASD. Moreover, there have not been adequate mechanisms for transferring practice-based knowledge from the information-rich areas such as Canada and the United States of America to researchers and service providers in the South to inform their own policy and practice. We argue, in agreement with Salmon and Clarren [122] for a globalization approach—interconnectedness and interdependence across the different regions of the world to share their experiences and knowledge in supporting multi-directional capacity building in FASD prevention and management.

## 5. Strengths and Limitations

One of the strengths of this study is that we aimed to include studies on the prevention and management interventions across the life span. In addition, thematic analysis was used to analyze the data (using a framework developed in accordance with the aim of the study), which allowed interventions to be categorized as prevention interventions and management interventions. 

The results of this review should be considered in light of the following limitations. A weakness is that this current scoping review selected studies that are published only in English from the selected databases. Some of the studies selected were limited by sample size and others were pilot studies, which required to be tested with larger samples. This review only focused on evidence published from 2007 to 2017 as was stipulated in the study objective. As is with other systematized reviews with a large number of articles, it takes some time to screen all the full data, classify the strength of the evidence, extract and analyze the data and then prepare the manuscript for submission. To avoid disruptions through these processes, we decided to focus our scoping review up to 2017. Inherent to reviews addressing the effectiveness of a program, intervention or policy is the issue of how to report efficacy. In this review, it was difficult to standardize the measure or assessment of effectiveness due to the variation of study designs used in the selected articles. For instance, we found that it was difficult to compare the effectiveness of a program obtained through a pre-post study finding to that obtained from an RCT in a comparable manner.

## 6. Conclusions

Despite FASD being preventable, its prevalence remains high around the world. Therefore, more efforts should be geared toward prevention. FASD is a lifelong disability with various educational, medical and social issues facing individuals affected by the disorders, however, there is a paucity of interventions for adolescents and adults. The lack of appropriate interventions for these age groups promotes the development of secondary disabilities. There is also a paucity of interventions developed and implemented in countries with a higher prevalence of FASD such as Europe and Africa. Therefore, there is a great need to share and implement context/culturally appropriate interventions for the prevention and management of FASD.

## Figures and Tables

**Figure 1 ijerph-16-02244-f001:**
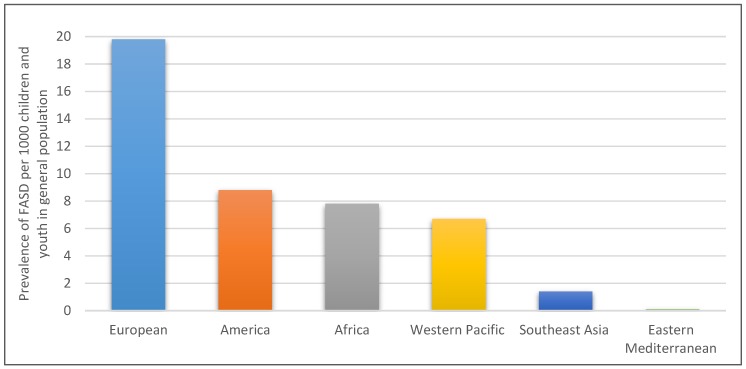
Prevalence of fetal alcohol spectrum disorder (FASD) by region.

**Figure 2 ijerph-16-02244-f002:**
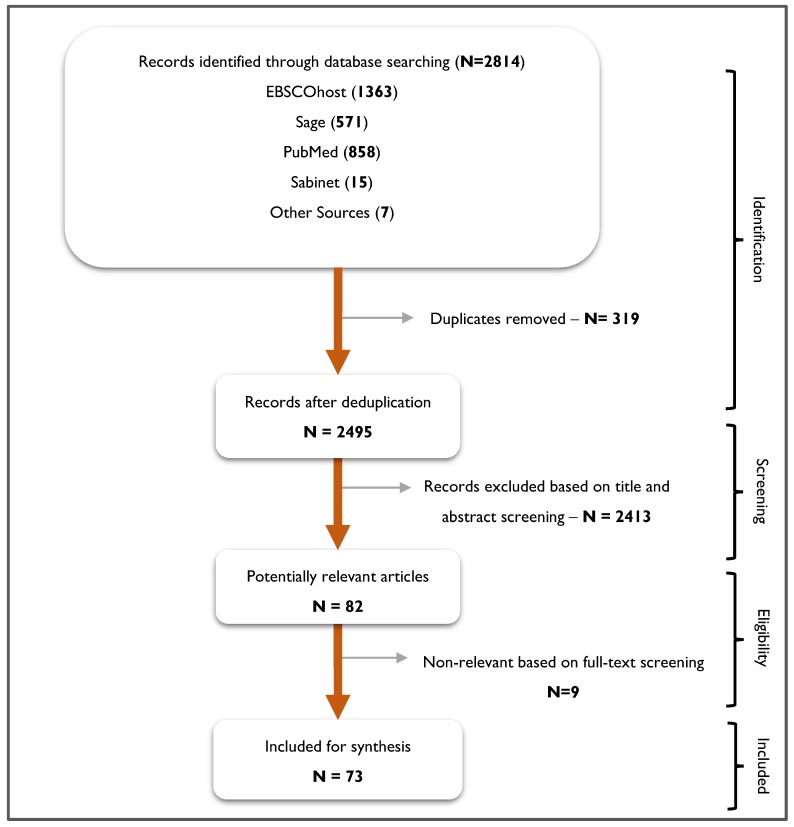
A flow chart for the selection process.

**Figure 3 ijerph-16-02244-f003:**
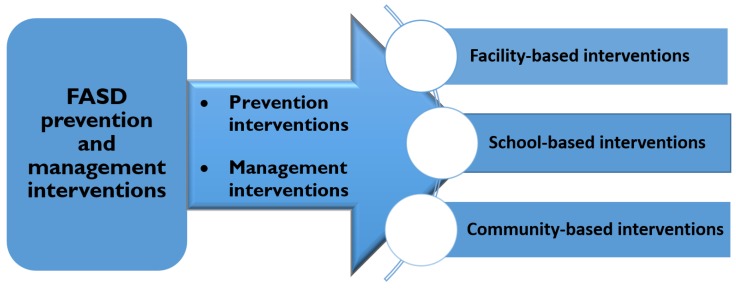
Heuristic classification framework applied to summarize the data.

**Figure 4 ijerph-16-02244-f004:**
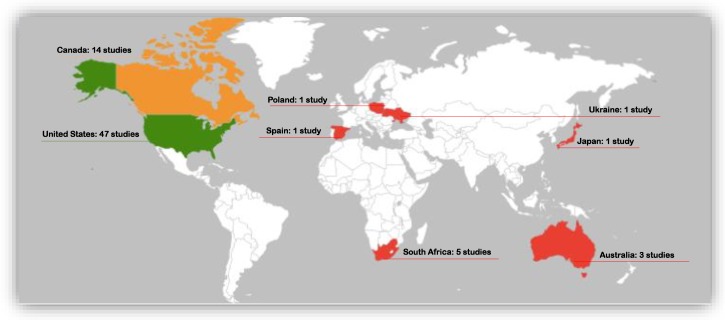
A global representation of the prevention and management intervention studies by countries.

**Table 1 ijerph-16-02244-t001:** Characteristics and criteria for PICO (Population, Intervention, Comparison, and Outcome).

Characteristic	Criteria
Population	Individuals with FASD, young people, adult, women, and children
Intervention	Any strategy aimed at preventing or managing FASD
Comparison	Individuals who do not receive interventions
Outcome	Effective in preventing or managing FASD

**Table 2 ijerph-16-02244-t002:** Quality assessment components and ratings for Effective Public Health Project Instrument.

Components	Strong	Moderate	Weak
Selection bias	Very likely to be representative of the target population and >80% participation rate	Somewhat likely to be representative of the target population and 60% to 79% participation rate	Not likely to be representative (i.e., self-referred), <60% participation rate or not stated
Design	Randomized controlled trial and controlled clinical trial	Cohort analytic, case-control, cohort, or an interrupted time series	All other designs or designs not stated
Cofounders	Controlled for at least 80% of confounders	Controlled for 60% to 79% of confounders	Confounders not controlled for, or not stated
Blinding	Blinding of outcome assessor and study participants to intervention status and/or research question	Blinding of either outcome assessor or study participants or blinding is not described	Outcome assessor and study participants are aware of intervention status and/or research question
Data collection methods	Tools are valid and reliable	Tools are valid but have not been shown to be reliable	No evidence of validity or reliability
Withdrawals and dropouts	Follow-up rate of >80% of participants	Follow-up rate of 60% to 79% of participants	Follow-up rate of <60% of participants or withdrawals and dropouts not described

**Table 3 ijerph-16-02244-t003:** Characteristics of the studies included in the review.

Characteristics	Number of Studies
**Research approach**	
Quantitative	72
Mixed method	1
**Study design**	
Mixed method	1
Descriptive longitudinal study	1
Prospective	3
Survey	2
Dichotomized control trial	2
Retrospective case analysis	4
Pre-post-test	13
Non-randomized control trial	2
Randomized control trial	40
Quasi-experimental	2
Case study	2
Cohort analytic	1
**Type of intervention**	
Prevention	32
Management	41
**Study setting (continent)**	
North America	61
Africa	5
Europe	3
Australia	3
Asia	1

**Table 4 ijerph-16-02244-t004:** Themes generated from the prevention interventions.

Prevention Interventions			
Nature of Interventions	Numbers of Studies and Citation	Interventions	Outcomes of Interventions
Facility-based	Joya et al. [43]	Single-session motivational interview for pregnant women	Decreased alcohol consumption during pregnancy
	Velasquez et al. [44]	Dual-focused approach (motivation interviewing on alcohol and contraception)	Reduced the risk for alcohol-exposed pregnancy (AEP) by increasing the effective use of contraception and decreasing alcohol use
	Payne et al. [45]	Educational resources on FASD for Health professionals	Increased practitioners’ knowledge of FASD
	Mwansa-Kambafwile et al. [13]	Training course on FASD capacity building for service providers	Built service providers’ capacity to prevent and manage women at risk for AEP
	Ondersma et al. [46]	Computer-Delivered Screening and Brief Intervention for pregnant women	Demonstrated efficacy for favorable birth outcome
	Delrahim-Howlett et al. [47]	Web-based alcohol assessment and personalized feedback for women of reproductive age	Reduced number of risky alcohol consumption
	Tzilos et al. [48]	Brief Computer-Delivered Intervention for pregnant women	Significantly decreased alcohol use
	Hutton et al. [49]	Project CHOICES intervention for women	Decreased the AEP risk in the through effective use of contraception and decrease alcohol use
	Osterman et al. [50]	Single-session of motivational interviewing (MI) for women	MI was not found effective in decreasing alcohol use
	Osterman and Dyehouse [51]	Motivational interview intervention for pregnant women	MI was not found effective in decreasing alcohol use
School-based/education-based	LaChausse [52]	Multimedia presentation on FASD for high school students	Increased the students’ knowledge of FASD
	Boulter [53]	Peer-delivered educational presentation on FASD for youth	Increased youths’ knowledge about FASD
	Toyama and Sudo [54]	Tailored leaflet educational intervention	Improved knowledge of pregnant women re FASD
Community-based	Hanson et al. [55]	Preconception prevention program (motivational interview techniques) for at-risk women	Reduced the risk for AEP by increasing contraception use
	Farrell-Carnahan et al. [56]	One-session motivational interview for non–treatment-seeking community women	Decreased the AEP risk in the community by increasing contraception use and decreasing alcohol use
	Hanson et al. [11]	Motivational interview-based intervention for women	Modified self-reported drinking and contraception behavior positively
	O’Connor and Whaley [10]	Brief intervention (10–15 min counseling sessions) for pregnant women	Promoted abstinence from alcohol by increasing motivation to change unhealthy behavior
	Letourneau et al. [57]	Mail-based prevention program for at-risk women	Reduced the risk for AEP by increasing the effective use of contraception
	Tenkku et al. [58]	Web-based intervention using tailored motivational messaging for women	Reduced the risk of an AEP by increasing the effective use of contraception and decreasing alcohol use
	De Vries et al. [12]	Case management intervention for heavy drinking pregnant women	Helped women to stop drinking and reduced the risk of FASD
	Montag et al. [59]	Targeted Screening, Brief Intervention, and Referral to Treatment (SBIRT) intervention for women	Decreased risky drinking behavior and vulnerability to AEP
	May et al. [60]	Case management intervention for heavy drinking pregnant women	Reduced maternal alcohol drinking at critical times
	France et al. [61]	Threat-based and self-efficacy based message on alcohol for women	Promoted women’s intentions to abstain from alcohol
	Wilton et al. [62]	Telephone-based brief intervention (counseling sessions)	Reduced the risk of an AEP by increasing contraception use and decreasing alcohol use
	Rasmussen et al. [63]	Mentorship program for at-risk women	Reduced the risk of an AEP by increasing contraception use
	Floyd et al. [64]	Brief motivational intervention for women	Reduced the risk of an AEP by increasing contraception use and decreasing alcohol use
	Chersich et al. [9]	Universal intervention (highlighting FASD using local media and health promotion talks at health facilities)	Reduced the prevalence of FASD by increasing knowledge of harms of maternal drinking
	Dresser et al. [65]	Training program on FASD for alcohol-server	Reduced serving of alcohol to pregnant women
	Hanson et al. [66]	Media campaign on FASD for women	Increased knowledge of FASD and decreased actual drinking
	May et al. [67]	Case management intervention for women	Helped women to stop drinking and reduced the risk of FASD
	Ingersoll et al. [68]	Pre-conceptional motivational interviewing interventions for women	Decreased DDD (drinks per drinking day), ineffective contraception rate and AEP risk
	Russell et al. [69]	Project CHOICES intervention for youth and adult	Demonstrated increased effectiveness of birth control use and decreased use and abuse of alcohol

**Table 5 ijerph-16-02244-t005:** Themes generated from the management interventions.

Management Interventions			
Nature of Interventions	Numbers of Studies and Citation	Interventions	Outcomes of Interventions
Facility-based	Nguyen et al. [70]	Use of choline supplement for children	Did not improve cognitive performance
	Zarnegar et al. [71]	Use of neuro-developmentally informed intervention for children	Improved developmental deficit in several domains
	Wozniak et al. [72]	Use of choline supplement for children	Improved neurocognitive functioning
	Wozniak et al. [73]	Use of choline supplement for children	Has potential to improve neurocognitive functioning
	Wilczynski et al. [74]	Use of sensory integration (SI) therapy for children	Improved gross motor function
	Yazdani et al. [75]	An early intervention program for mother and young children	Mitigated cognitive deficit
	Kable et al. [76]	Use of choline supplement with multivitamin/mineral for children	Impacted brain development positively
	Connolly et al. [77]	Applied behavior analysis (ABA)-based verbal behavior treatment program for children	Showed rapid skill acquisition in communication adaptive emotional/behavioral functioning
	Nash et al. [78]	Individual designated therapy room for children	Ameliorated executive functioning deficits
	Soh et al. [79]	Sensory integration and cognitive behavioral training	Improved self-regulation skills and brain development
	Keiver et al. [80]	Physical activity program for children with FASD	Showed differences in cortisol levels in children with FASD compared to Controls
	Nash et al. [81]	Alert® Program for Self-Regulation for behavioral regulation for children	Improved functional integrity in the neural circuitry for behavioral regulation
	Doig et al. [82]	Attention-deficit/hyperactivity disorder (ADHD) treatment for children	Inattention may be less responsive to ADHD medication
School-based/education-based	Coles et al. [83]	Educational intervention for specific learning and behavior need for children	Improved both mathematical skill and behavior
	Kerns et al. [84]	Use of computerized attention training for children	Improved cognitive performance
	Jirikowic et al. [85]	Intervention to increase compliance with motor function in children	Improved sensory adaptation, balance and motor performance
	Adnams et al. [86]	Classroom language and literacy intervention for children	Improved cognitive in targeted brain areas
	Kable et al. [87]	The educational intervention focused on behavior mathematical functions for children	Remediated mathematical deficits
	Kable et al. [88]	Computerized instruction for children	Improved self-regulation
	McCoy et al. [89]	Sensorimotor Training for children	Increased postural sway velocity
	Keil et al. [90]	Social skills intervention	Improved deficits in social information-processing
	Schonfeld et al. [91]	Social-skills and play-therapy	Improved social skills and reduced problem behaviors
	Wells et al. [92]	Group therapy intervention for foster and adoptive caregivers and children	Improved the executive functioning and emotional problem-solving
	O’Connor et al. [93]	Alcohol reduction intervention for adolescents with FASD	Reduced and prevented alcohol use and some secondary disabilities
	Kerns et al. [94]	Game-based process specific intervention for children	Improved cognitive development
	Coles et al. [95]	Computer game for a metacognitive control strategy for children	Reduced disruptive behaviors
	Coles et al. [96]	Virtual reality game of fire safety and street safety for children	Showed significantly better knowledge of the game to which they were exposed
	Loomes et al. [97]	Rehearsal training on working memory span of children	Showed improvement in digit span in children receiving rehearsal training
	Clark et al. [98]	Professional development program for teachers	Provided support for the effectiveness of the professional development program for teachers
Community-based	Leenaars et al. [99]	Families intervention program	Assisted families to cope with stress
	O’Connor et al. [100]	Social-skills and play-therapy for children and parents	Improved knowledge of appropriate social skills, and parent-reported social skills
	Petrenko et al. [101]	Tailored intervention for children and caregivers	Improved child self-regulation and caregiver behavior
	Reid et al. [102]	Home-based program for high-risk, vulnerable families	Improved self-regulatory skills
	Petrenko et al. [103]	Program targeting key risk and protective factors for children and families	Showed promising result for the prevention of secondary conditions and improves the family adaptation
	Millians and Coles [104]	A program developed to address neurocognitive mathematical impairments for children	Remediated learning problems
	Kable et al. [105]	Mathematical skills intervention for children	Improved mathematical skill
	Pomeroy and Parrish [106]	Online training on FASD for Court Appointed Special Advocates	Improved FASD knowledge
	Hanlon-Dearman et al. [107]	Use of community home-based attachment intervention for caregivers	Improved communication of needs
	Kable et al. [108]	Caregiver Education and Training for Behavioral Regulation for children	Improved caregivers’ knowledge of FASD
	Pelech et al. [109]	Intervention to enhance practice at child welfare and improve placement stability	Significant decline in number of placement changes
	Denys et al. [110]	Step by Step mentor program for parents	Significant reduction in client’s needs and significant increase in client’s goals

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
