# Peer review of "The Distribution of Available Prevention and Management Interventions for Fetal Alcohol Spectrum Disorder (2007 to 2017): Implications for Collaborative Actions"

_ijerph, 2019, doi:10.3390/ijerph16122244_

Round 1

Reviewer 1 Report

see attached or below

The present manuscript,  A Global Distribution of Available Prevention and Management Interventions for Fetal Alcohol Spectrum Disorder (2007 to 2017): Implications for Collaborative Actions, addresses an important topic in a timely review of prevention and management interventions for fetal alcohol spectrum disorders (FASD).  As awareness of FASD increases and diagnosis becomes more accessible, the need for interventions intensifies. The goal of this manuscript was to identify, classify, and map by country, various existing interventions described in publications over a ten-year period. The authors utilize well-accepted criteria for evaluation and briefly report intervention descriptions and outcomes in a readable, organized fashion. Conclusions include calls for interventions in areas where they are lacking (e.g. management interventions for adults with FASD) and increased global cooperation to reach underserved populations and maximize benefits of accumulated knowledge.

General comments:

1.       Because this review is limited to a specific and recent time period, consider discussing impactful intervention literature prior to the present sample (e.g. some of the seminal work by Phil May) in the introduction or discussion

2.       The inclusion/exclusion criteria could be tightened up to explain why some material is not included (e.g. do you choose among studies regarding the same intervention? Soh 2015 re Alert). No problem to limit scope but should be specified

3.       Inherent to reviews is the issue of how to report efficacy. The quality assessments in “Additional file 1” are an excellent approach but it is left to the reader to interpret their effect on findings reported in “Additional file 2”. Should we confidently report a significant drop in alcohol consumption when there is a >50% loss to follow-up? Should a pre-post study finding be reported in the same way as an RCT? All but two interventions are reported as effective (Montag 2015 and Nguyen 2016) and these are two of the strongest studies in terms of quality. Results for Montag 2015 could alternatively be described as alcohol consumption and risk for AEP decreased significantly from baseline in all groups and were maintained through the 6-month follow-up (as it would have been had it been a pre-post study). Even better would be to include Montag AJPH 2015;105(8):1572-6 where the intervention resulted in a significantly greater reduction in alcohol consumption compared to assessment alone among depressed women.

4.       The places where tailoring to diverse communities and the importance of culture is mentioned are appreciated and could be expanded

Specific comments:

Introduction

1.       Pg 1, line 33 - …individuals exposed to alcohol prenatally…

2.       Pg 2, line 42 – This is a shifting landscape, perhaps say “…results in diagnosable FASD…”. As our assessments become more sensitive, we may be able to identify a greater proportion

3.       Pg 2, line65 – sadly, not just “stigmatization of alcoholics” but stigmatization of birth mothers whether alcoholics or not (Corrigan 2018)

4.       Pg 2, line 70 – “broad” diverse?

5.       Pg 2, line 76 – “poor” inadequate?

6.       Pg 2, line 76 – consider being broader in your definition of reviews here and including more reviews that your readers may appreciate (e.g. Petrenko and Alto 2016)

7.       Pg 3, line 83-84 – unclear – more precisely targeted interventions can be effective – think you mean that a series of effective interventions geared to different stages may be optimal?

8.       Pg 3, line 85-86 – please clarify this gap

Materials and methods

9.       Pg 4, - expand inclusion/exclusion criteria

10.   Pg 6, lines 147-148 – please clarify Juni comment and how this affects your manuscript

Results

11.   Table 3 title – “study” studies

12.   Pg 10, line 214 – only 2?

13.   Pg 12, line 245 – be specific as “cognitive performance” is a broad term and other studies show benefit within this category

14.   Pg 12, line 257 – “and it showed” resulted in

Discussion

15.   Pg 13, line 327-329 include in inclusion/exclusion criteria or results

16.   Pg 13, line 332 – “plurality” paucity?

17.   Pg 13, line 333 – secondary disabilities can more easily…

18.   Pg 14, line 338 – include culturally-appropriate too

19.   Pg 14, lines 340-350 – consider discussing the need for tailoring to maximize efficacy; interventions work best when motivations and methods reflect the communities’ world view and culture

20.   Pg 14, line 383 – “rollout” modified for implementation?

21.   Pg 14, lines 384-386 – good!

22.   Pg 15, line 404 – across the lifespan…  Also, not really true since you didn’t find interventions geared to more than certain ages. Consider encouraging publication of ongoing efforts. Quan 2018 provides guidance for developing community-based interventions for adults

Author Response

Reviewer comments

The present manuscript,  A Global Distribution of Available Prevention and Management Interventions for Fetal Alcohol Spectrum Disorder (2007 to 2017): Implications for Collaborative Actions, addresses an important topic in a timely review of prevention and management interventions for fetal alcohol spectrum disorders (FASD).  As awareness of FASD increases and diagnosis becomes more accessible, the need for interventions intensifies. The goal of this manuscript was to identify, classify, and map by country, various existing interventions described in publications over a ten-year period. The authors utilize well-accepted criteria for evaluation and briefly report intervention descriptions and outcomes in a readable, organized fashion. Conclusions include calls for interventions in areas where they are lacking (e.g. management interventions for adults with FASD) and increased global cooperation to reach underserved populations and maximize benefits of accumulated knowledge.

Response to Reviewer 1

General comments:

Query - 1. Because this review is limited to a specific and recent time period, consider discussing impactful intervention literature prior to the present sample (e.g. some of the seminal work by Phil May) in the introduction or discussion.

Response:  Thanks for pointing this out. We have added reviews that included intervention studies out our specific period.

Action: Although the conclusions drawn from the reviews were limited by the poor methodological quality and paucity of studies as indicated by the authors [14, 15]. Psychological and educational interventions and other interventions delivered during antennal care have potentials to increase abstinence and reduce alcohol consumption among pregnant women [14, 15].  Reid et al. [22] showed that there is growing evidence supporting the effectiveness of FASD management interventions in improving outcomes for early to middle childhood. However, there is a dearth of management interventions for individuals with FASD beyond early and middle childhood [22] suggesting inadequate management plan for these individuals. Peadon et al. [23] reported that pharmacological and non-pharmacological management interventions have showed some benefit among children with FASD. While Paley and O’Connor [25] confirmed that behavioral interventions showed immediate post-intervention effects, they did not ascertain the long-term follow-up outcomes.

Query - 2.       The inclusion/exclusion criteria could be tightened up to explain why some material is not included (e.g. do you choose among studies regarding the same intervention? Soh 2015 re Alert). No problem to limit scope but should be specified

Response: I have tightened up the inclusion/exclusion. Also, some of the articles might have been missed because of the type of databases and journals subscribed for full-text articles by our University in a particular period.

Action: We have done the search again to include this article, however, any article that it is not included after this new search maybe due to the reason mentioned earlier.

Query - 3.       Inherent to reviews is the issue of how to report efficacy. The quality assessments in “Additional file 1” are an excellent approach but it is left to the reader to interpret their effect on findings reported in “Additional file 2”. Should we confidently report a significant drop in alcohol consumption when there is a >50% loss to follow-up? Should a pre-post study finding be reported in the same way as an RCT? All but two interventions are reported as effective (Montag 2015 and Nguyen 2016) and these are two of the strongest studies in terms of quality. Results for Montag 2015 could alternatively be described as alcohol consumption and risk for AEP decreased significantly from baseline in all groups and were maintained through the 6-month follow-up (as it would have been had it been a pre-post study). Even better would be to include Montag AJPH 2015;105(8):1572-6 where the intervention resulted in a significantly greater reduction in alcohol consumption compared to assessment alone among depressed women.

Response:  Thanks for this. We have added below.

Action: Inherent to reviews addressing the effectiveness of a program, intervention or policy is the issue of how to report efficacy. In this review, it was difficult to standardize the measure or assessment of effectiveness due to the variation of study designs used in the selected articles. For instance, we found that it was difficult to compare the effectiveness of a program obtained through a pre-post study finding to that obtained from an RCT in a comparable manner.

Query - 4.  The places where tailoring to diverse communities and the importance of culture is mentioned are appreciated and could be expanded

Response: We added below

Action: Our finding aligns with that of Petrenko and Alto [27] on the need to address the barriers to the implementation of FASD interventions, especially cultural barriers as highlighted in their work. We support the call for collaboration among researchers, cultural experts and local stakeholders to facilitate implementation, which will enhance the effectiveness of interventions and promotes sustainability. We agreed with their suggestion on the use of purveyors as cultural liaisons between researchers and local stakeholders to facilitate buy-in from the latter and the community at large. The importance of the above is to encourage tailored interventions to maximize efficacy. In addition, we believed that interventions work best when the motivations and methods reflect the communities’ world view and culture.

Specific comments:

Introduction

Query - 1.       Pg 1, line 33 - …individuals exposed to alcohol prenatally…

Response: Thanks for your suggestion

Action: “alcohol prenatally…” has been added.

Query - 2.       Pg 2, line 42 – This is a shifting landscape, perhaps say “…results in diagnosable FASD…”. As our assessments become more sensitive, we may be able to identify a greater proportion

Response: Thanks for your suggestion

Action: “diagnosable” has been added

Query - 3.       Pg 2, line65 – sadly, not just “stigmatization of alcoholics” but stigmatization of birth mothers whether alcoholics or not (Corrigan 2018)

Response: Thanks for your suggestion

Action: “birth mother” has been added

Query - 4.       Pg 2, line 70 – “broad” diverse?

Response: Thanks for suggestion this

Action: “diverse” has been added

Query - 5.       Pg 2, line 76 – “poor” inadequate?

Response: Thanks for suggestion this

Action: “poor” has been change to “inadequate’’

Query - 6.       Pg 2, line 76 – consider being broader in your definition of reviews here and including more reviews that your readers may appreciate (e.g. Petrenko and Alto 2016)

Response: Thanks for suggesting this

Action: We have included more review. “Systematic reviews have been published on interventions for the prevention [14, 15] and management of FASD [22, 23] and other literature reviews for management interventions [25–29].’’

Query - 7.    Pg 3, line 83-84 – unclear – more precisely targeted interventions can be effective – think you mean that a series of effective interventions geared to different stages may be optimal?

Response: Thanks for point this out. The sentence has been made clearer

Action: FASD interventions need to be diverse and include prevention and both pharmacological and non-pharmacological management approaches across the life span as the negative effects of prenatal alcohol exposure can manifest across all ages.

Query - 8.  Pg 3, line 85-86 – please clarify this gap

Materials and methods

Response: Thanks for pointing this out. We have added below.

Action: From the reviews above, we observed that there is a lack of a review for prevention of FASD outside those who are pregnant, planning a pregnancy and indigenous community. In addition, we discovered the lack of review for both pharmacological and non-pharmacological interventions that assessed methodological rigor across the life span.

Query - 9.       Pg 4, - expand inclusion/exclusion criteria

Response: It has been expanded.

Inclusion criteria:

·         The interventions (both pharmacological and non-pharmacological) must aim at preventing or improving the outcome of prenatal exposure to alcohol;

·         Articles published in the English Language;

·         Articles published from 2007 to 2017 (we chose this period to provide current information to help inform policy development);

·         The target population must be women, young people, and individuals with FASD;

·         All the types (randomized controlled trials {RCT}, quasi RCT, non-randomized controlled trials and cohort studies with pre- and post-intervention 

·         Interventions targeting any age group.

Exclusion criteria:

·         Studies that do not report on the effectiveness or promising outcome of the interventions;

·         Animal studies;

·         Other systematic, scoping and literature reviews.

·         Unpublished prevention and management interventions

·         Articles published before 2007 and after 2017

Query - 10.   Pg 6, lines 147-148 – please clarify Juni comment and how this affects your manuscript

Results

Response: Thanks for pointing this out. We have added the below.

Action: Herein, the overall rating of the quality of the studies assessed was not done.

Query - 11.   Table 3 title – “study” studies

Response: Thanks for observing this

Action:  “study” has been changed to “studies”

Query - 12.   Pg 10, line 214 – only 2?

Response: Thanks for observing this

Action: “only’’ has been included

Query - 13.   Pg 12, line 245 – be specific as “cognitive performance” is a broad term and other studies show benefit within this category

Response: Thanks for your suggestion

Action:

Query - 14.   Pg 12, line 257 – “and it showed” resulted in

Response: Thanks for your suggestion

Action: “and it showed” has been changed to ‘’resulted in’’

Discussion

Query - 15.   Pg 13, line 327-329 include in inclusion/exclusion criteria or results

Response: Thanks for pointing this out.

Action: We have included this in the result and it was also modified to suite discussion.

Query - 16.   Pg 13, line 332 – “plurality” paucity?

Response: Thanks for observing this

Action: “plurality” has been changed to ‘’paucity’’

Query - 17.   Pg 13, line 333 – secondary disabilities can more easily…

Response: Thanks for observing this

Action: ‘’more’’ has been added

Query - 18.   Pg 14, line 338 – include culturally-appropriate too

Response: Thanks for suggesting this

Action: ‘’culturally-appropriate’’ has been added

Query - 19.   Pg 14, lines 340-350 – consider discussing the need for tailoring to maximize efficacy; interventions work best when motivations and methods reflect the communities’ world view and culture

Response: Thanks for your suggestion. We have added below.

Action: The importance of the above is to encourage tailored interventions to maximize efficacy. In addition, we believed that interventions work best when the motivations and methods reflect the communities’ world view and culture

Query - 20.   Pg 14, line 383 – “rollout” modified for implementation?

Response: Thanks for suggesting this

Action: “rollout” has been change to ‘’modified for implementation’’

Query - 21.   Pg 14, lines 384-386 – good!

Response:  Thank you

Action: No action

Query - 22.   Pg 15, line 404 – across the lifespan…  Also, not really true since you didn’t find interventions geared to more than certain ages. Consider encouraging publication of ongoing efforts. Quan 2018 provides guidance for developing community-based interventions for adults

Response: Thanks for pointing this out. We may have included interventions up to certain age, however, the interventions for other age categories would have been included if found. I have rephrased the statements as below.

Action: One of the strengths of this study is that we aimed to include studies on the prevention and management interventions across the lifespan. We also added this statement ‘’ Furthermore, we encourage publication of ongoing effort prevent and management FASD across the lifespan”

Reviewer 2 Report

The authors have undertaken a scoping review of the available prevention and management approaches for FASD that are currently available. I have provided detailed suggestions below. 

Abstract: 

Line 17 instead of saying various databases – would list the databases that were searched 

Introduction: 

line 48 – could combine this paragraph with the next paragraph as both small and on the same topic 

Line 65 – instead of saying alcoholics could say “people with alcohol use disorders or alcohol addiction” 

Line 66 – change wording of the sentence > lack of skills by service providers to discuss alcohol use with women who are pregnant

Line 67 – could include here something about the importance of contraception and preventing unplanned pregnancies – as if just focus on preventing alcohol use once people know they are pregnant – will not ultimately prevent FASD 

Line 78 – what have previous reviews of prevention interventions reported? It was my understanding that a number of previous reviews reported that many current approaches were not effective at changing alcohol use behaviour. 

Line 83 to 86 – confusing sentences – switching between interventions for those affected and prevention interventions. Also would be good to provide more information about what is meant by ‘all categories.’

Line 93 – not clear what the gaps are the current review are filling e.g. Could include that you are providing an update to the literature – given previous reviews were published a number of years ago – also that given that only including publications from 2007 – could say that providing a scoping review of more recent literature to help inform current policy development 

Methods: 

Inclusion criteria – would be good to explain why the year 2007 was chosen 

Was an updated search completed before submission? Given that it is now 2019 would be good to do an updated search and include any additional papers. 

Flow chart –  process for identification is different to what is usually done – is this in line with a process required for scoping review? E.g. numbers are usually combine from all databases 1st and then duplicates removed at the start. Title and abstract are usually used to screen 1st and then full text subsequently. 

Results: 

Need to include summary of quality assessment in the results section – usually present this before the summary of study findings 

Table 4 – would be good to include 1st author details in the table for each row – so readers can easily see the results of each included study 

Would be good to have a more critical review of the prevention studies – a number of studies don’t report significant changes in alcohol use or between the interventions tested – would be good to know in what way risk for AEP was reduced e.g. was it a change in contraceptive behaviour or alcohol use behaviour or both? 

Table 5 – row 4 – would re-word outcome – not clear what this means – did choline lead to behaviour improvements? 

Row 6 – same – not sure what mitigated damages means – improved behaviour or other type of functioning? 

row 7 how was brain development impacted – negatively or positively? And in what way 

Row 8 – Not sure what verbal behaviour treatment means? rapid skill acquisition in what?  

Row 9 - wording change suggestion– could say improved or ameliorated executive functioning deficits or impairments

Discussion: 

Line 327 – suggest to change the wording of this sentence – currently sounds as though only including studies of those up to 18 

Line 341 – change to scant 

Line 383 – change to rolled out 

Conclusions – need to include information about prevention – focused on interventions for management 

Spelling – intervention > interventions in this section 

Author Response

Reviewer 2 comments

The authors have undertaken a scoping review of the available prevention and management approaches for FASD that are currently available. I have provided detailed suggestions below. 

Response to reviewer 2 comments.

Abstract:

Query - Line 17 instead of saying various databases – would list the databases that were searched

Response: Thanks for point this out. We do not listed the databases because of the word counts for the abstract was limited to 200 words. However, the sentence was rephrased to below.

Action:  We searched some selected databases with predefined terms.

Introduction:

Query - line 48 – could combine this paragraph with the next paragraph as both small and on the same topic

Response: Thanks for suggestion this

Action: The paragraphs have been combined together

Query - Line 65 – instead of saying alcoholics could say “people with alcohol use disorders or alcohol addiction”

Response: Thanks for your suggestion

Action: “alcoholics” has been changed “people with alcohol use disorders

Query - Line 66 – change wording of the sentence > lack of skills by service providers to discuss alcohol use with women who are pregnant

Response: Thanks for point this out.

Action: The wordings of the sentence has been changed to “lack of skills by service providers to discuss alcohol use with women who are pregnant”

Query - Line 67 – could include here something about the importance of contraception and preventing unplanned pregnancies – as if just focus on preventing alcohol use once people know they are pregnant – will not ultimately prevent FASD

Response: Thanks for pointing this out

Action: We have added “unplanned pregnancy through the use of contraceptives”

Query - Line 78 – what have previous reviews of prevention interventions reported? It was my understanding that a number of previous reviews reported that many current approaches were not effective at changing alcohol use behaviour.

Response: Thanks for pointing this out. We have added below

Action: Although the conclusions drawn from the reviews were limited by the poor methodological quality and paucity of studies as indicated by the authors [14, 15]. Psychological and educational interventions and other interventions delivered during antennal care have potentials to increase abstinence and reduce alcohol consumption among pregnant women [14, 15].

Query - Line 83 to 86 – confusing sentences – switching between interventions for those affected and prevention interventions. Also would be good to provide more information about what is meant by ‘all categories.’

Response: Thanks for pointing this out. We have structure the sentences as below

Action: FASD interventions need to be diverse and include prevention and both pharmacological and non-pharmacological for management across the life span as the negative effects of prenatal alcohol exposure can manifest across all ages. Therefore, the need to address the gap of a study exploring the prevention interventions for all women and the management interventions across life span necessitates this current scoping review.

Query - Line 93 – not clear what the gaps are the current review are filling e.g. Could include that you are providing an update to the literature – given previous reviews were published a number of years ago – also that given that only including publications from 2007 – could say that providing a scoping review of more recent literature to help inform current policy development

Response: Thanks for your suggestion. We added below.

Action: From the reviews above, we observed that there is a lack of a review for prevention of FASD outside those who are pregnant, planning a pregnancy and indigenous community. In addition, we discovered the lack of review for both pharmacological and non-pharmacological interventions that assessed methodological rigor across the life span. FASD interventions need to be diverse and include prevention and both pharmacological and non-pharmacological management approaches across the life span as the negative effects of prenatal alcohol exposure can manifest across all ages. Therefore, the need to address the gap of a study exploring the prevention interventions for all women and the management interventions across life span necessitates this current scoping review.Thirdly, to update previous reviews and lastly to provide current information to help inform policy development.

To this end, we aimed to conduct a scoping review that will serve four purposes. First, the review will help to identify the prevention and management interventions of FASD reported in the literature across the life span to address the above-mentioned gaps in interventions for FASD. Secondly, we sought to map the concentration of these interventions across the globe vis-à-vis the burden of FASD in the different regions. Thirdly, to update previous reviews and lastly to provide current information to help inform policy development [31–34].

Methods:

Query - Inclusion criteria – would be good to explain why the year 2007 was chosen

Response: Thanks for pointing this out. We have added below statement

Action: “we chose this period to provide current information to help inform policy development.’’

Query - Was an updated search completed before submission? Given that it is now 2019 would be good to do an updated search and include any additional papers.

Response: This review only focused on evidence published from 2007 to 2017 as is stipulated in the study objective. As is with other systematized reviews with a large number of articles, it takes some time to screen all the full data, classify the strength of the evidence, extract and analyze the data and then prepare the manuscript for submission. To avoid disruptions through these processes, we decided to focus our scoping review up to 2017.

Action: No action was taken

Query - Flow chart – process for identification is different to what is usually done – is this in line with a process required for scoping review? E.g. numbers are usually combine from all databases 1st and then duplicates removed at the start. Title and abstract are usually used to screen 1st and then full text subsequently.

Response: Thanks for pointing out this.

Action: It has been corrected.

Results:

Query - Need to include summary of quality assessment in the results section – usually present this before the summary of study findings

Response: Thanks for bringing this up. The part for quality assessment in the discussion has been moved to under methods.

Action: It has been moved

Query - Table 4 – would be good to include 1st author details in the table for each row – so readers can easily see the results of each included study

Response: Thanks for this.

Action: It has been included.

Query - Would be good to have a more critical review of the prevention studies – a number of studies don’t report significant changes in alcohol use or between the interventions tested – would be good to know in what way risk for AEP was reduced e.g. was it a change in contraceptive behaviour or alcohol use behaviour or both?

Response: Thanks for mentioning this. It has been clarified

Action: It has been clarified

Query - Table 5 – row 4 – would re-word outcome – not clear what this means – did choline lead to behaviour improvements?

Response: Thanks for this. We have change the out to below.

Action: Has potential to improve neurocognitive functioning

Query - Row 6 – same – not sure what mitigated damages means – improved behaviour or other type of functioning?

Response: Thanks for this. We have change the statement to below

Action: Mitigated cognitive deficit

Query - Row 7 how was brain development impacted – negatively or positively? And in what way

Response: Thanks for pointing this out

Action: ‘’positively’’ has been added

Query - Row 8 – Not sure what verbal behaviour treatment means? rapid skill acquisition in what? 

Response: Thanks for pointing this out. We have clarify the above below.

Action: Applied behavior analysis (ABA)–based Verbal Behavior treatment program for children, ‘’Showed rapid skill acquisition in communication adaptive emotional/behavioral functioning’’

Query - Row 9 - wording change suggestion– could say improved or ameliorated executive functioning deficits or impairments

Response: Thanks for suggesting this

Action: The wording has been changed to ‘’ameliorated executive functioning deficits’’

Discussion:

Query - Line 327 – suggest to change the wording of this sentence – currently sounds as though only including studies of those up to 18

Response: Thanks for pointing this out

Action: We changed the wording to ‘’Although, this review targeted interventions across the life span for both prevention and management interventions, we could not find studies for management interventions that include participants above 18 years of age for this period (2007 – 2017). Most of the management interventions reported targeted children.’’

Query - Line 341 – change to scant

Response: Thanks for your suggestion

Action: It has been changed to ‘’scant’’

Query - Line 383 – change to rolled out

Response: Thanks for your suggestion

Action: ‘’roll out’’ has been changed to ‘’modified for implementation’’

Query - Conclusions – need to include information about prevention – focused on interventions for management

Response: Thanks for pointing this out

Action: We added this “Despite FASD is preventable the prevalence remains high around the world. Therefore, more efforts should be geared toward prevention.”  

Query - Spelling – intervention > interventions in this section

Response: Thanks for your suggestion

Action: ‘’intervention’’ has been changed to ‘’interventions’’
